

# Leveraging 20 Years of Remote Sensing to Characterize Surface Phytoplankton Seasonality and Long-Term Trends in Lake Tanganyika

François Toussaint[1], Alice Alonso[1], Marnik Vanclooster[1]

[1]Earth and Life Institute – Environmental Sciences (ELIE), Faculty of Biosciences Engineering, UCLouvain, Croix du Sud 2, L7.05.02, BE – 1348 Louvain-la-Neuve, Belgium

*Correspondence to*: François Toussaint (francois.toussaint@uclouvain.com)

**Abstract.** Lake Tanganyika, the world's second-largest freshwater lake by volume, is a vital resource for millions in East Africa, providing water, food, and economic opportunities while supporting exceptional biodiversity.

Chlorophyll-a concentration (Chl-a) is a key indicator of phytoplankton biomass and primary productivity, and thus a proxy for the health of aquatic ecosystems. In Lake Tanganyika, Chl-a is known to display strong spatiotemporal horizontal variability with an exceptionally low annual mean and wide ranges of concentrations compared to other tropical or temperate great lakes. This variability is influenced by the lake's hydrodynamic cycle driven by air temperature and wind seasonality. Phytoplankton biomass is suspected to be decreasing due to a strengthening of water column stratification induced by 15 climate change. However, the particular spatiotemporal variability and trends in phytoplankton biomass have never been examined using a lake-wide, temporally continuous long-term record. This study bridges this gap by analyzing satellite remote sensing-derived Chl-a data from the ESA Climate Change Initiative Lakes dataset across the entire surface of Lake Tanganyika over a 20-year period. It offers insight into the Chl-a dynamics with an unprecedented timespan and spatial coverage.

The analysis reveals distinct seasonal patterns in Chl-a concentrations, with shallow regions (depth <170 m) maintaining high levels year-round, while deeper areas exhibit pronounced seasonality tightly linked to known wind patterns. To further explore these spatial differences in seasonal dynamics, the study identifies seven clusters of co-varying Chl-a concentrations, each displaying distinct seasonal behaviours that reflect the lake's hydrodynamic cycle. Long-term trends indicate a decline in Chl-a concentrations of -9% per decade in deep regions, suggesting decreasing primary productivity. However, this 25 overall decline is nuanced by monthly patterns. In deep regions, the low Chl-a concentrations, mainly observed between November and April, tend to decrease over time at rates between -5 to -15% per decade when averaged over entire clusters. In contrast high Chl-a values recorded during the most productive months, from August to October, show increasing trends up to 25%. Nearly all shallow areas, meanwhile, display year-round increases up to 35% across the Chl-a distribution, with particularly sharp rises in extreme values.

The findings underscore the complexity of Lake Tanganyika's Chl-a dynamics. The observed trends may have significant consequences for the lake's trophic structure and the communities dependent on its resources. Further research is needed to elucidate the underlying drivers of these changes and to assess their broader ecological and socio-economic impacts.



# 1 Introduction

Lakes play a fundamental role in global biogeochemical cycles and provide essential ecosystem services, serving as crucial

water resources for millions of people (Wetzel, 2001). However, these fragile ecosystems face increasing pressure from climate change, pollution and other stressors, which drive profound changes in their physical, chemical, and biological properties (Brönmark & Hansson, 2002). Given their sensitivity to environmental shifts, lakes are often considered sentinels of change, rapidly responding to alterations in temperature, precipitation, and human-induced disturbances (Adrian et al., 2009; Williamson et al., 2009). Continuous monitoring of water quality and ecosystem health is essential to better

understand these ecosystems, for assessing the transformations they are undergoing and guiding adaptive management efforts.

One important indicator of lake ecosystem health is phytoplankton biomass, as it is a marker of primary productivity and responds to environmental changes (Xu et al., 2001). Variations in phytoplankton biomass can signal shifts in trophic state, alterations of ecosystem dynamics, or direct anthropogenic pressures such as water pollution. Since phytoplankton form the

base of the aquatic food web, fluctuations in their abundance directly influence higher trophic levels and therefore the whole ecosystem (Wetzel, 2001). Measurements of phytoplankton concentration are typically made at discrete locations, either via boat campaigns or from easily accessible buoys (Kutser, 2009; Lombard et al., 2019). However, in large bodies of water, such methods may not adequately represent the overall water quality across the entire surface. Additionally, temporal availability can present challenges, as these methods are often limited in sampling frequency. In recent decades, the use of

optical satellite remote sensing to estimate an indicator of phytoplankton biomass has emerged as a powerful tool, providing broad spatial coverage and continuous temporal monitoring across large water bodies (Gordon & Wang, 1994; Hu & Campbell, 2014; Martin, 2014; McClain et al., 2006; Moore et al., 2014; Moses et al., 2009; Neil et al., 2019). It involves estimating chlorophyll-a (Chl-a) concentration by utilizing its distinctive absorption peaks in the blue and red bands (Martin, 2014).

Lake Tanganyika, located in East Africa, is the second most voluminous freshwater lake in the world. It spans four countries, Tanzania, the Democratic Republic of the Congo, Burundi, and Rwanda. It is an essential resource for millions of people living along its shores, who rely on the lake's fish as their primary source of animal protein but also for water and economic opportunities (Bulengela, 2024; Mölsä et al., 2002; Niyongabo et al., 2024; Paffen et al., 1997). The lake is characterized by its unique ecosystem, which supports a wide range of endemic species, particularly in its fish communities (Coulter, 1991).

This ecosystem is increasingly exposed to a multitude of threats as climate change poses significant challenges, exacerbated by human activities (Plisnier et al., 2018). Extreme high water levels in recent years have had severe consequences, causing floods and population displacements (Gbetkom et al., 2024; Papa et al., 2023). Physical parameters of the lake are changing such as water temperature that increases along with air temperature, causing a strengthening of vertical stratification (O'Reilly et al., 2003; Verburg et al., 2003). This increased stratification is expected to reduce nutrient upwelling, thereby



limiting phytoplankton growth, the key driver of the lake's food web and fisheries. Reduction in vertical mixing has caused the oxycline to rises, restricting habitats to shallower areas (Cohen et al., 2016; Van Bocxlaer et al., 2012).

Lake Tanganyika's water quality and ecosystem dynamics have been the focus of numerous studies, many of them focusing on primary production and phytoplankton concentrations (Bergamino et al., 2007, 2010; Corman et al., 2010; Descy et al., 2005; Hecky & Fee, 1981a; Hecky & Kling, 1981; Langenberg et al., 2002; Stenuite et al., 2007). Field campaigns have usually focused on specific regions of the lake, often nearshore and during limited time periods (Plisnier, et al., 2023). Phytoplankton concentrations in the lake are generally low but they exhibit high spatial and temporal variability, making it challenging to characterize their lake-wide dynamics with sparse in situ measurements. Bergamino et al. (2010) used MODIS remote sensing data to estimate annual primary production over the entire lake surface based on three years of observations. Their study was limited to a short time series and did not characterize seasonal variability or assess long-term trends in phytoplankton dynamics. No study to date has provided a comprehensive, long-term assessment of phytoplankton dynamics across the entire lake surface, preventing a full understanding of spatial patterns, seasonal cycles, and long-term trends.

This study aims to analyse the spatiotemporal variability of surface Chl-a concentrations across the entire surface of Lake Tanganyika, with particular attention to regional differences in seasonal dynamics. We also seek to assess long-term trends over the past two decades and examine changes in the statistical distribution of Chl-a, with an emphasis on extreme values. Finally, we aim to compare Chl-a trends between deep and shallow zones of the lake to identify potential differences in their responses to environmental and climatic drivers.

## 2 Material and Methods

### 2.1 Study Site

Lake Tanganyika is the largest African Great Lake, located between -3.6°N and -8.8°N and 29.2 and 31.2°E (Figure 1). It is 650 km long, averaging 50 km of width along a North-West to South-East axis and has an area of more than 32 600 km2. It comprises three main basins, with the deepest point reaching 1 400 meters located in the central basin.

The region's climate is characterized by a cool and wet season from September to April and a warmer, drier season from May to August, with air temperatures generally ranging between 25°C and 29°C and an average annual precipitation of approximately 1,000 mm. Strong southeast winds peak in July-August before weakening in September as the Intertropical Convergence Zone shifts (Nicholson, 2000), while lighter northerly winds dominate the rest of the year (Coulter, 1991).

Surface water temperatures vary seasonally between 24 to 28 °C while deep-water temperatures remain stable at 23.45 °C. Lake Tanganyika is highly stratified and meromictic, meaning its water column never fully mixes (Coulter, 1991). Seasonal monsoon winds dominate the mixing processes. During the cooler dry season (May–August), strong southeast winds tilt the thermocline, causing it to rise closer to the surface in the south. This disruption of stratification in the south favors upwellings, which enhance nutrient cycling and promote phytoplankton growth. Simultaneously, surface waters are pushed





northward, causing downwelling in the northern basin. As winds relax during the wet season transition (September–October), stratification returns in the south while secondary upwellings occur in the north. Internal seiching, a lake-wide oscillation of the thermocline, becomes prominent, driving mixing and primary production especially when its amplitude

reaches surface waters (Corman et al., 2010; Coulter, 1991; Langenberg et al., 2002, 2003; Naithani et al., 2002; Plisnier et al., 2023; Plisnier et al., 1999; Plisnier & Coenen, 2001).

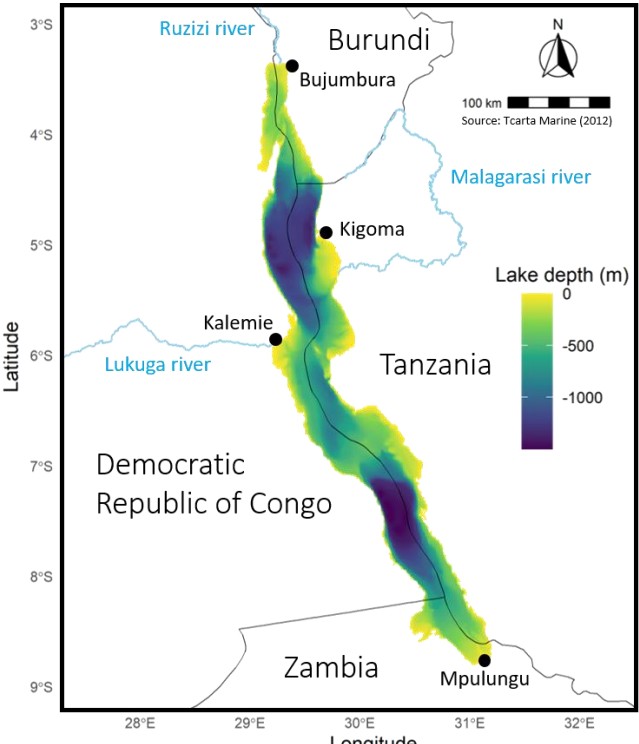

**Figure 1: Map of Lake Tanganyika bathymetry, surrounding countries, major towns, and key rivers (Inflow: Ruzizi, Malagarasi; Outflow: Lukuga).**

Lake Tanganyika's water is remarkably clear and nutrient levels at the surface are generally low. Phytoplankton

concentration show great spatiotemporal variability depending on the nutrient availability in the euphotic zone. The seasonality of phytoplankton concentration, often measured in terms of Chl-a levels or estimates of primary production, depends on the hydrodynamic cycle and winds seasonality. Phytoplankton biomass is located between depths from 0 to 40 m with maxima commonly found between 0-20 m (Descy et al., 2010; Salonen et al., 1999; Vuorio et al., 2003). Bergamino et al. (2010) used remote sensing data from 2002 to 2005 to estimate primary productivity over the whole lake's surface based

on MODIS data. They computed an average of $646 \pm 142$ mg C.m$^{-2}$ per day. They also showed the remarkable differences between some coastal zones around the lake in shallow areas showing values above 800 mg C.m$^{-2}$ per day and the rest of the lake where primary production was lower. The lake's stratification regime has indeed confined the distribution of benthic



organisms to a narrow band of biodiversity around the lake within the upper, oxygen-rich water layer (Van Bocxlaer et al., 2012).

Lake Tanganyika has experienced significant warming, with recent temperature increases unprecedented in the past 1500 years (Tierney et al., 2010). Since 1913, the upper water column has warmed at approximately 0.1 °C per decade (Kraemer et al., 2015; O'Reilly et al., 2003), with higher rates in the north due to weaker vertical mixing. Seasonal variations exist, with slower warming during the dry season (0.08 °C per decade) and faster warming near the shore (Verburg & Hecky, 2009). This warming trend is expected to increase the stability of the water column and reduce vertical mixing (O'Reilly et

al., 2003; Tierney et al., 2010; Verburg et al., 2003; Verburg & Hecky, 2009).

Direct measurements of primary production have been too intermittent to determine whether the changes in the lake have caused a decrease in productivity (Macintyre, 2012). O'Reilly et al. (2003) suggested a 20% decline in primary productivity and a 30% reduction in fish yields based on carbon isotope records from sediment cores. Reduced silica demand by diatoms further indicates a shift toward a more oligotrophic system (Verburg et al., 2003; Verburg & Hecky, 2009). Stenuite et al.

(2007) reported daily mean and daily minimum primary production below previous estimates while Bergamino et al. (2010) estimated primary productivity in 2003 using remote sensing data and found a 15% decrease compared to the estimate by Hecky & Fee (1981), though the studies used different methodologies.

An increase in water temperature is also expected to reduce the mixing and oxycline depths, reducing ecosystem habitats (Cohen et al., 2016; O'Reilly et al., 2003; Verburg et al., 2003). Indeed, if vertical mixing is reduced, the oxygen consumed

by metabolic respiration cannot be replenished through mixing, leading to its depletion. This should cause a narrowing of the ring of benthic biodiversity in the lake. Concurrently, water clarity has improved as phytoplankton levels have declined. This could enhance benthic primary production relative to that of pelagic phytoplankton (Van Bocxlaer et al., 2012). All these environmental changes are coupled with observations of declining fish populations even before the era of commercial fishing (Cohen et al., 2016).

**2.2 Dataset**

We used the ESA Climate Change Initiative Lakes dataset (Lakes_cci Version 2.1) (Carrea et al., 2024).This dataset contains Essential Climate Variables (ECV) such as lake water level and extent, lake surface water temperature (LSWT), Chl-a and water turbidity for over 2000 lakes worldwide. The Chl-a coverage spanned from 2002 to 2022, with a theoretical daily temporal resolution, that may vary due to cloud cover, and a spatial resolution of 1/120th of a degree of latitude. This

roughly corresponds to a spatial resolution of 1 km near the equator. Data from different satellite products (MERIS on Envisat, MODIS on Terra and Aqua, and OLCI on Sentinel-3) were harmonized to produce a temporally consistent Chl-a dataset. The Lakes_cci dataset also provides uncertainties layers associated with each ECV. For Chl-a, 90% of the dataset had an associated fractional uncertainty between 39 and 60%. Most lakes, including Tanganyika, show data gaps during the 2012 to 2016 period due to validation issues. Only 48 lakes in the dataset contain Chl-a measurements from MODIS during

this period. Among the African Great Lakes, only Lake Victoria has continuous coverage of Chl-a.



## 2.3 Data Interpolation Method

Due to cloud cover and satellite revisit times, the dataset contains gaps, with some days missing data entirely or having incomplete coverage of the lake. To address this, we applied an interpolation method to fill in missing data and ensure continuity in the analysis. The Data Interpolating Empirical Orthogonal Functions (DINEOF) technique was used for spatio-temporal interpolation of Chl-a and LSWT data (Alvera-Azcárate et al., 2005; Beckers & Rixen, 2003). The method consists in iteratively decomposing and reconstructing the dataset in a set of empirical orthogonal functions (EOF), expressed as spatial and temporal modes. This process is repeated until the root mean square error, computed on a spatially coherent set of validation pixels, converges to a stable value. These validation pixels were selected as contiguous areas of 10 000 pixels from the 15 images with the least cloud coverage in the dataset. We found 7 to be the optimal number of EOF modes that minimizes the error function. We applied this interpolation procedure solely to images containing less than 95% missing values, as recommended by the method. Other images were not altered and remained in the dataset in their original state.

## 2.4 Seasonality of Chl-a Concentrations

To capture the seasonal dynamics of surface Chl-a concentrations, we first generated monthly median maps of Chl-a by computing the median of all values observed during each month at each location. These maps illustrate the general spatial patterns of Chl-a across the lake. However, they cannot be used to display the short-term and interannual variability in Chl-a.

## 2.5 Spatial Clustering for Improved Characterization of Chl-a variability

To address this limitation and provide a more detailed characterization of the temporal variability of Chl-a in Lake Tanganyika, we partitioned the lake into regions with co-varying Chl-a concentrations rather than analysing the lake as a whole. To do the partitioning, we used k-means clustering (Macqueen, 1967) with the seven spatial empirical orthogonal function modes previously computed using the DINEOF method as input. The optimal number of spatial clusters, determined using the silhouette value (Rousseeuw, 1987), was found to be 7. To characterize the temporal variability of Chl-a, we computed two complementary time series for each cluster. The first captures the interannual variability of Chl-a, calculated as the 10th, 25th, 50th, 75th, and 90th quantiles of all observations for each month between January 2002 and December 2022. The second time series represents the seasonal cycle of Chl-a and is calculated as the 10th, 25th, 50th, 75th, and 90th quantiles of all Chl-a estimates within each cluster for each day of the year, pooling data across all years in the time series. This means that each day's statistics represent interannual daily percentiles, derived from the corresponding day across all years in the dataset. We had first applied a five-day rolling mean to the time series of each pixel over the entire period from 2002 to 2022. This smoothing method was selected to reduce the impact of outliers, minimize the influence of missing data while preserving short-term variability, following the approach of Laliberté & Larouche (2023).



### 2.6 Overall and Monthly Absolute and Relative Trends in Chl-a Concentrations

Chl-a trends were independently calculated for each pixel across Lake Tanganyika using daily time-step data from the lakes_cci dataset for the period 2002 to 2022. Before conducting the trend analysis, we first removed the seasonal cycle. To do this, we determined the interannual median Chl-a value for each pixel $i$ on each day of the year (DOY), averaging across all years in the time series. These seasonal cycles were then smoothed using a 5-day rolling mean. Finally, Chl-a anomalies were calculated by subtracting the smoothed seasonal values from the observed Chl-a concentrations following Eq. (1), ensuring that the trend analysis focused on long-term changes rather than recurring seasonal fluctuations.

$$Chl\text{-}a\ A_{i,t} = Chl\text{-}a_{i,t} - \widetilde{Chl\text{-}a}_{i,DOY} , \tag{1}$$

where $Chl\text{-}a_{i,t}$ is the observed Chl-a concentration for pixel $i$ on day $t$, with $i = 1,\dots, N$ (the total number of pixels covering the lake) and $t$ representing the day number within the period from January 1, 2002, to December 31, 2022. $\widetilde{Chl\text{-}a}_{i,DOY}$ is the interannual median Chl-a concentration for pixel $i$ on the corresponding DOY, averaged across all years, and smoothed using a 5-day rolling mean. $Chl\text{-}a\ A_{i,t}$ represents the deviation of the observed Chl-a from the expected seasonal median value.

Overall Trends in Chl-a were calculated using the Sen's slope estimator $\tilde{\beta}$ following Eq. (2), expressed in mg.m$^{-3}$ per decade, alongside a Mann-Kendall test for significance. The Sen's slope, defined as the median of all pairwise slopes in the time series (Sen, 1968), provides a robust estimate of trend magnitude.

$$\tilde{\beta}_i = median\left(\frac{Chl\text{-}a\ A_{i,t_j} - Chl\text{-}a\ A_{i,t_k}}{t_j - t_k}\right) \ for\ all\ t_j > t_k , \tag{2}$$

where $Chl\text{-}a\ A_{i,t_j}$ and $Chl\text{-}a\ A_{i,t_k}$ are the Chl-a anomalies for pixel $i$ at times $t_j$ and $t_k$, respectively.

Firstly, overall absolute trends were computed for each pixel using all available observations and are expressed as changes in surface Chl-a concentration per decade. Overall relative trends were then computed by dividing the overall absolute trends by the median Chl-a concentration at each pixel following Eq. (3), providing a relative measure of change, expressed in percentage by decade.

$$relative\ \tilde{\beta}_i = \frac{\tilde{\beta}_i}{\widetilde{Chl\text{-}a\ A_i}} , \tag{3}$$

where $\widetilde{Chl\text{-}a\ A_i}$ is the median Chl-a concentration anomaly at pixel $i$ over the study period.

To capture intra-annual variations of these trends, monthly relative trends were also estimated using only the observations for each specific month of the year.





## 2.7 Shifts in Chl-a Distribution: Quantile-Based Trend Analysis

In addition to analysing overall and monthly trends per pixel, we assessed if there was any shift in yearly statistical distribution of Chl-a over time. To achieve this, we computed trends for a set of yearly quantile values, consisting of those at 2.5% intervals along with the uppermost quantiles at 98%, 98.5%, 99%, 99.5%, and 99.9%. Recognizing that lake-wide

trends might obscure local variability, we conducted this analysis within the seven previously defined clusters of similar temporal Chl-a variability. For each pixel, we calculated all yearly quantile values and determined trends from 2002 to 2022 using the Sen's slope estimator following Eq. (4). Relative trends were then calculated by dividing each yearly quantile trend by its interannual mean, yielding rates of change. Finally, we spatially aggregated the results by calculating the median rate of change for each quantile within each cluster following Eq. (5).

$$\tilde{\beta}_{q,i} = median\left(\frac{q_{i,y_j} - q_{i,y_k}}{y_j - y_k}\right), \tag{4}$$

$$relative\ \tilde{\beta}_{q,C} = median\left(\frac{\tilde{\beta}_{q,i}}{\bar{q}_i}\right) \forall i\ \in C\ , \tag{5}$$

where $\hat{\beta}_{q,i}$ is the Sen's slope estimator for the $q^{th}$ quantile, calculated as the median of pairwise slopes between yearly quantile values $q_{i,y_j}$ and $q_{i,y_k}$ for pixel $i$ over years $y_j$ and $y_k$ between 2002 and 2022. $relative\ \tilde{\beta}_{q,C}$ is the median rate of change for the $q^{th}$ quantile within a specific cluster $C$ , calculated based on all pixels $i$ within this cluster. $\bar{q}_i$ is the

interannual mean of $q^{th}$ quantile at pixel $i$ across all years. This analysis was conducted separately for pixels in deep zones and deep in shallow zones, as these two areas are influenced by distinct processes affecting primary productivity.

## 3 Results

### 3.1 Seasonality of Chl-a Concentrations

The maps of interannual monthly median surface Chl-a shown in Figure 2 highlight strong differences between some coastal

zones and the rest of the lake. These coastal zones usually extend a few kilometres offshore and are relatively shallow, with depths generally not exceeding 250 m (Figure 1). They exhibit consistently high to very high median concentrations throughout the year, above 3 mg.m$^{-3}$ in some areas. In contrast, open waters with greater depths and deep nearshore zones exhibit a more pronounced Chl-a seasonality, displaying low median levels outside the productive season between January and April, below 0.3 mg.m$^{-3}$.

Shallow zones with year-round high Chl-a levels are scattered around the lake. The northern zone, off the coast of Bujumbura and the Bujumbura Rural Province, feature an extensive shallow area where depths of less than 100 meters stretch up to 6 kilometres offshore. It is characterized by consistently elevated Chl-a concentrations, which also extend further south along the gently sloped shores of Burundi. The Gulf of Burton, located south of Baraka on the north-western




coast, shares similarities with the waters off Bujumbura, featuring a large shallow zone with sustained high Chl-a levels
compared to deeper areas.

The coastal area south of Kigoma spans 700 km² at depths of less than 200 meters, including 400 km² shallower than 100
meters. It remains highly productive throughout the year. Some of the highest Chl-a concentrations in Lake Tanganyika are
observed at the Malagarasi river's estuary where a 40 km² area near the river mouth exhibits mean Chl-a levels exceeding 3
mg.m$^{-3}$. Other remarkably productive regions include the shallow zones near Kalemie and the southeastern Gulf in the DRC
as well as areas near Mpulungu in Zambia.

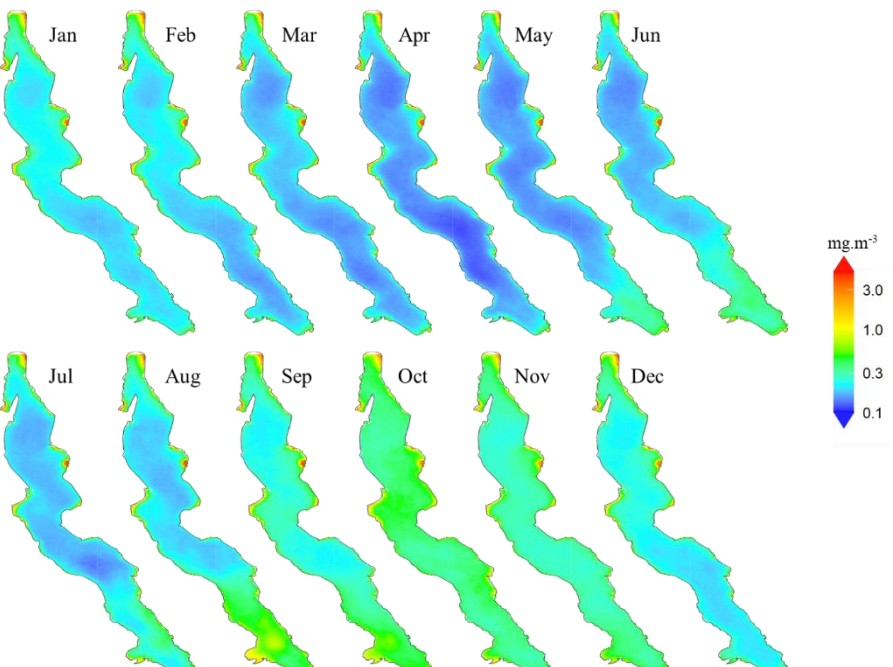

**Figure 2: Maps of interannual monthly median surface chlorophyll-a concentrations in Lake Tanganyika (2002-2022).**

By contrast, the Tanzanian coast north of Kigoma, where depth sharply increases offshore, shows Chl-a levels and
seasonality comparable to that of the pelagic waters at this latitude. Similar patterns are observed along the DRC coast, both
north of Kabimba and north of the border with Zambia, for example. Deep pelagic and coastal areas of the lake display
monthly median surface concentrations below 1 mg.m$^{-3}$ with the lowest median levels between 0.1 and 0.3 mg.m$^{-3}$ observed
in April. Chl-a levels start rising in the southern regions of the lake in April and this increase gradually progresses northward
in the following months, reaching the northern areas only by September. Peak concentrations are typically observed between
August and November in the south and between October and November in the north. From December to April, Chl-a
concentrations progressively decrease across the entire lake, starting from the south.



### 3.2 Spatial Clustering for Improved Characterization of Chl-a variability

Figure 3 illustrates the spatial aggregations into clusters of co-varying Chl-a concentrations that divide Lake Tanganyika from north to south into six relatively spatially coherent regions. Cluster 7 stands out as it groups pixels from shallow coastal regions across the lake, all characterized by consistently high Chl-a levels throughout the year.

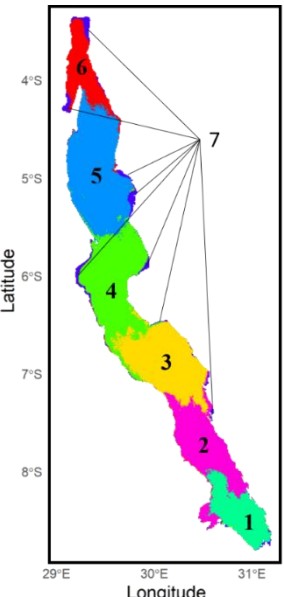

**Figure 3: Clustering of Lake Tanganyika into regions of co-varying chlorophyll-a concentrations with k-means clustering method.**

Chl-a observations within each cluster were aggregated to visualize both interannual variability and seasonal patterns. The left panel in Figure 4 highlights substantial interannual variability across all clusters. Clear seasonal patterns emerge in clusters 1 to 6, characterized by alternating periods of low and high productivity. However, the timing of peak Chl-a concentrations within the productive season varies from year to year, and multiple peaks may occur within a single year. The magnitude of these peaks also fluctuates greatly between years. In contrast, cluster 7, which represents the shallow coastal zones surrounding the lake, is characterized by persistently high Chl-a concentrations throughout the entire observation period and year-round, with no clear seasonal pattern, as shown in the right panel of Figure 4. Median daily Chl-a concentrations in this cluster consistently range around 1 mg.m$^{-3}$. Although they show remarkable interannual variability, the other clusters do show clear general seasonal patterns. From late April, median concentrations start to increase in clusters 1 and 2, at the very south of the lake. This increase persists until August and September, with median daily levels remaining below 1 mg.m$^{-3}$ during this period. The daily 75th and 90th percentile time series show recurrent peak values that indicate frequent phytoplankton blooms in these regions. At the very end of the dry season, high surface Chl-a levels are observed at the southern tip of Lake Tanganyika. Concentrations in clusters 1 and 2 then drop back to lower levels in September until mid-October. This coincides with the first peaks of Chl-a in the northernmost clusters 5 and 6, following a slow increase in concentrations that started in July. This northern Chl-a increase at the beginning of the wet season is followed by a gradual





southward shift of phytoplankton blooms across all clusters 5, 4, 3, and 2 over the following month. The magnitude of these blooms varies from year to year, both in spatial extent and in peak concentrations, but they consistently follow a southward progression. The September-October blooms in clusters 5, 4 and 3 are followed by a sharp decrease in the 75th and 90th quantiles time series, indicating a calm period with fewer extreme bloom events and a general decrease in concentrations towards the end of the wet season. Cluster 6 experiences blooms that gradually weaken and become less frequent until the end of the wet season. The two southernmost clusters reach their highest Chl-a values in October and November, before showing a decline in concentrations over the following months, with recurrent but less intense blooms.

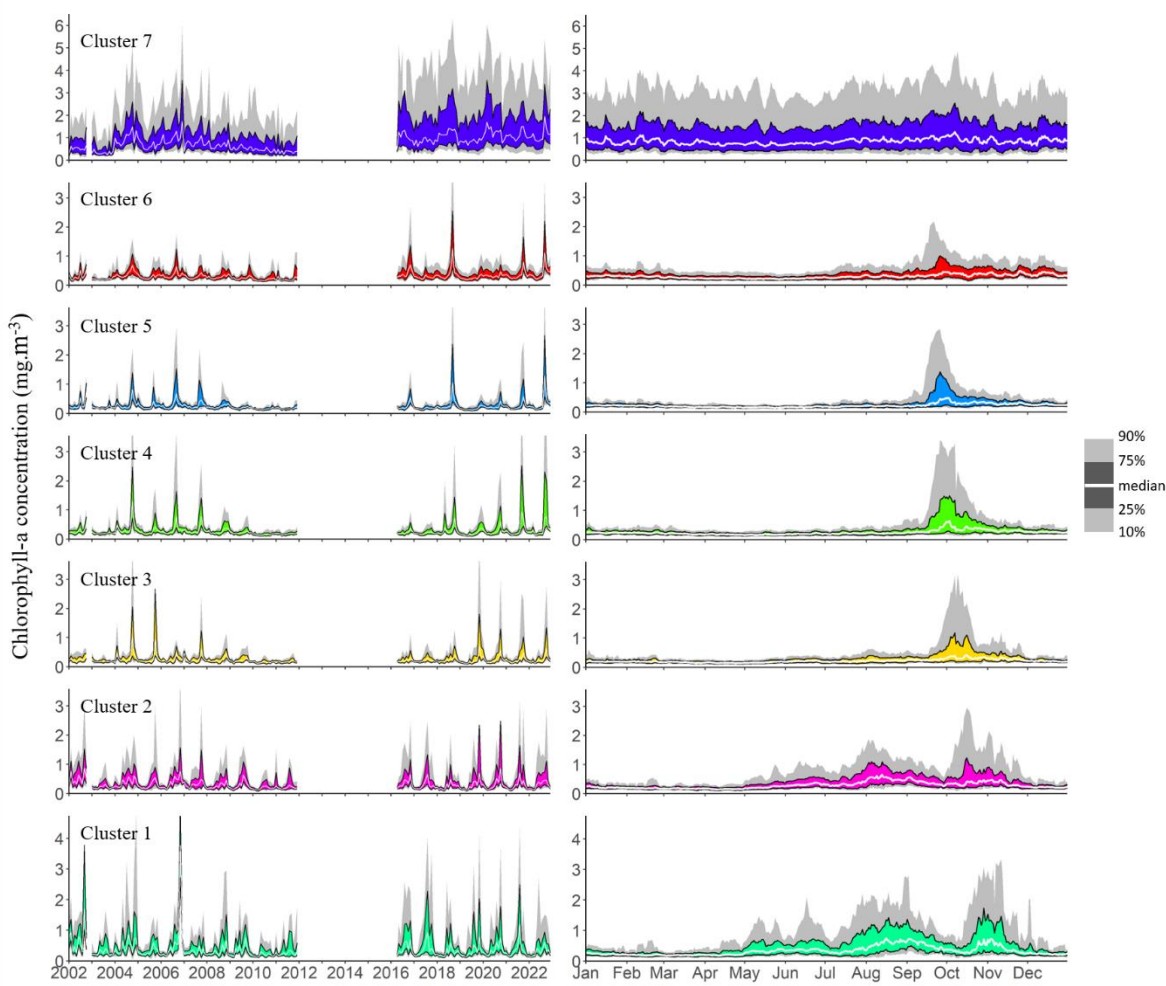

**Figure 4: Interannual and seasonal variations in chlorophyll-a concentrations, represented by the 10th, 25th, 50th, 75th, and 90th quantiles of all observations within each cluster. The left panel shows interannual variability based on monthly quantiles from 2002 to 2022, while the right panel depicts the seasonal cycle using daily quantiles pooled across the same period.**




## 3.2 Overall Patterns of Change in Surface Chl-a

Figure 5 shows maps of overall absolute and relative decadal trends in Chl-a concentrations in Lake Tanganyika and the corresponding p-values. A general decrease in Chl-a concentrations is observed across most of the lake's surface, except in shallow areas where concentrations remain high throughout the year. To examine how trends vary with depth, we computed the mean Chl-a trend for all pixels within the same depth range, using 10 m depth intervals. The results show positive mean trend values in areas with maximum depths up to 170 m, beyond which the average trends become negative. For subsequent analyses, this depth will serve as the boundary distinguishing the deep pelagic and coastal zones from the shallow coastal zones of the lake. Negative absolute trends found in deep regions are small, rarely exceeding -0.05 mg.m⁻³ per decade. They are generally statistically significant, as large areas show p-values below 1%. Although the negative trends indicate a slight decrease in Chl-a, they are found in regions where average Chl-a levels are already very low. When expressed relative to the interannual median Chl-a concentrations, these relative trends correspond to rates of change ranging from close to 0 down to -25% per decade, with an average relative trend of -9% per decade. The most severe relative trends in deep zones are found in the north basin and in the south of the lake.

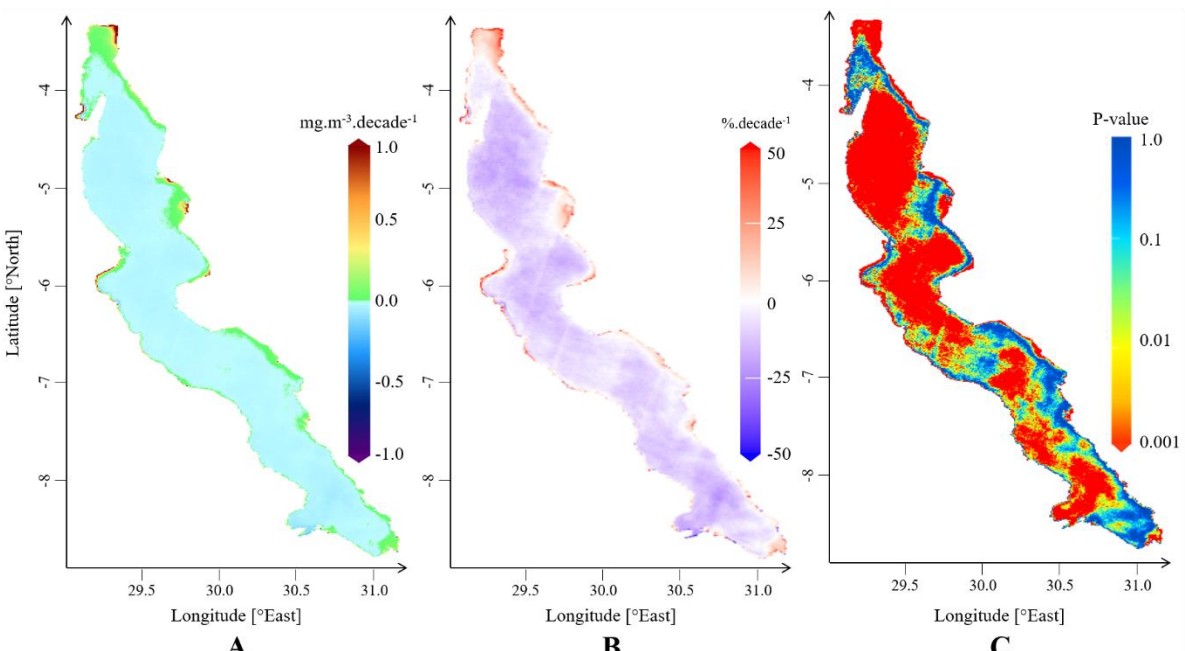

**Figure 5: Maps of overall trends in surface Chl-a in Lake Tanganyika, depicting absolute trends (A), relative trends computed with respect to pixel-specific median Chl-a concentrations (B) and corresponding p-values (C).**

Positive trends are observed across 13% of the lake's surface, mostly in shallow coastal areas that exhibit the highest year-round levels of Chl-a. These shallow zones with depth less than 170 m show a wide range of trend values compared to deeper areas, with both lower negative and higher positive absolute trends. Negative relative trends in shallow zones are





generally less pronounced than those in deep areas. 25% of trends in shallow areas exceed 10% per decade and 10% exceed 25% per decade, with some locations reaching increases of up to 50%. The maximum values are found near Bujumbura, Kigoma and Kalemie, at the estuary of the Malagarasi river and in the Gulf of Burton, where trends values exceed 1 mg.m⁻³ per decade.

**3.3 Monthly Dynamics of Surface Chl-a Trends**

Throughout the year, the lake's dynamics reveal a complex pattern of Chl-a trends. Figure 6 illustrates monthly relative trends in Chl-a concentrations, calculated based on monthly trends divided by the monthly median. Pelagic trends appear to closely align with Chl-a seasonality. In the deep areas of the lake, trends generally shift from predominantly decreasing between February and July to mostly increasing between August and October. From February to April, as Chl-a concentrations reach their lowest annual levels, starting in the south and progressing northwards as seen in Figure 2, negative Chl-a trends follow the same south-to-north pattern, extending across all deep regions of the lake by April. Relative trends show peak negative values of approximately -30% per decade, as seen in Figure 6, amounting to absolute trends up to -0.05 mg.m⁻³ per decade. Starting in May, trends in deep zones increase from the south. By August, Chl-a trends reach a first positive peak in southern clusters 1, 2, and 3, with relative increases ranging between 10 and 20% per decade and up to 30% per decade in the far south. In contrast, August trends show no statistical significance in the northern deep areas of the lake. In September, significant positive trends are found in the centre and northern part of the lake, reaching values of up to 50% per decade in some areas. October is characterized by widespread positive trends in surface Chl-a levels, with the highest values observed in the central and southern parts of the lake. From November to January, trend values become less pronounced and less significant, coinciding with the rainy season when the availability of optical data is at its lowest. Contrasting with the seasonality of pelagic Chl-a trends, shallow zones exhibit positive Chl-a trends throughout the year. These trends remain relatively stable in magnitude.



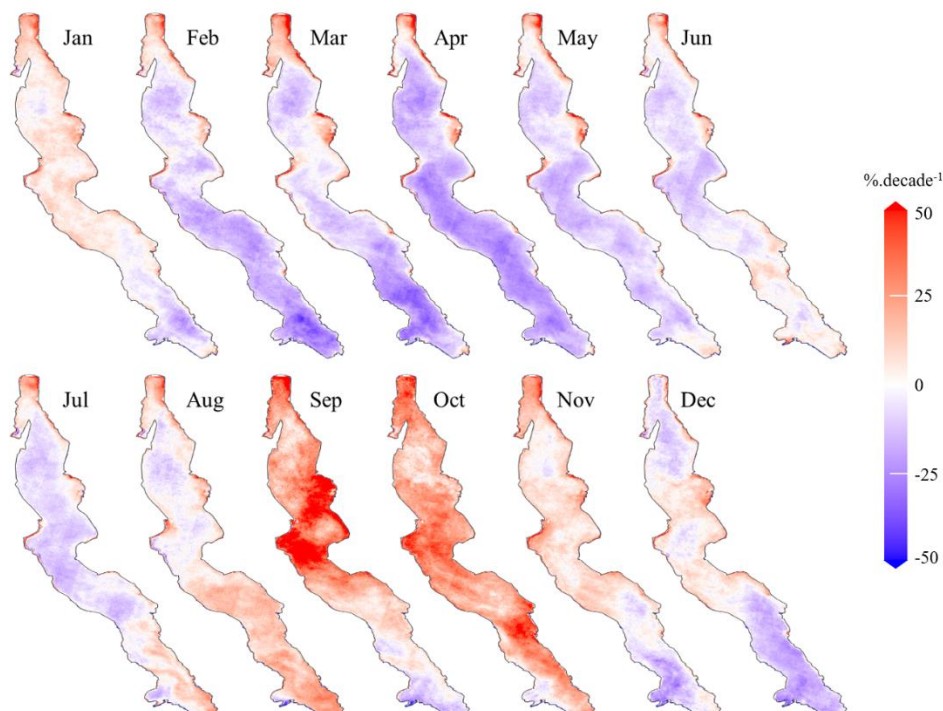

**Figure 6: Maps of monthly relative trends in Chl-a concentrations in Lake Tanganyika (2002-2022).**

## 3.4 Shifts in Chl-a Distribution: Quantile-Based Trend Analysis

To evaluate how the yearly distribution of Chl-a concentrations has evolved from 2002 to 2022, we analysed trends of yearly quantiles values for each pixel covering Lake Tanganyika. Figure 7 shows spatial median quantile trends by cluster across the yearly statistical distribution of Chl-a, with a distinct focus on deep and shallow regions. Overall, we see that nearly all clusters exhibit slightly positive changes in shallow areas for most of their distribution, with higher trends for the highest quantile values. In deep zones, most clusters exhibit negative changes for quantiles up to the 75th to 90th percentiles.

More specifically, shallow areas show the steepest increases in cluster 7, while cluster 2 exhibits decreasing values between -5 and -10% per decade across most of its distribution, below the 80th quantile. Clusters 1 and 4 display near-zero changes in their lower quantiles while clusters 3, 5 and 6 show modest increases of between 5 and 15% per decade. Notably, all clusters exhibit sharp increases in the highest quantiles, ranging from 10 to 30% per decade. For trends in deep areas, clusters 1 to 5 exhibit decreases for all quantiles below the 75th to the 90th percentiles, with values ranging from 0 to -15% per decade. In contrast, higher quantiles show sharp increases, ranging from 10 to 30% per decade. Cluster 6, located at the northern end of the lake, shows consistent increases around 5% per decade for the entire distribution below the 90th percentile. For both shallow and deep zones of the lake, this peaks in trends for the highest yearly quantiles could suggest more frequent extreme events, although statistical tests failed to detect any significant changes in the frequency or extent of major bloom events.




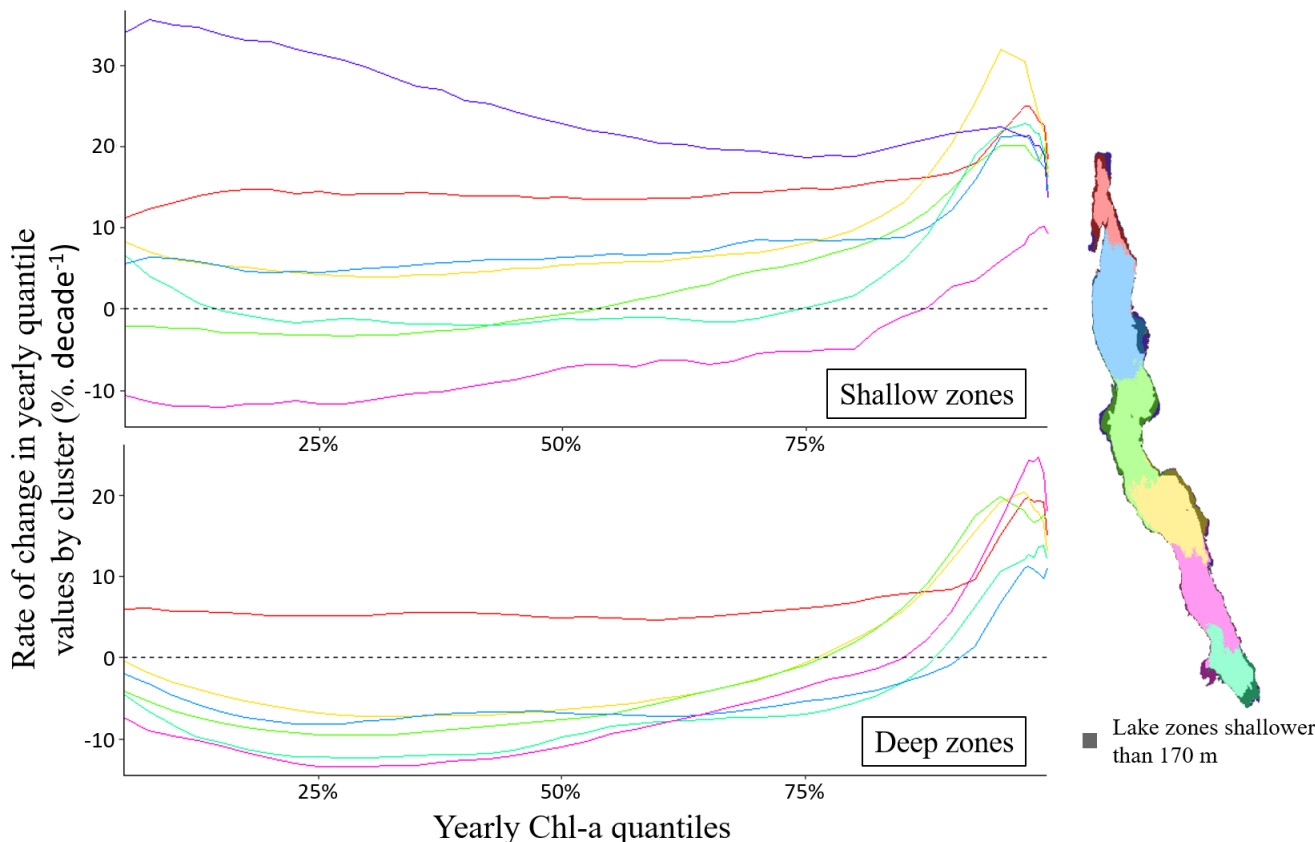

**Figure 7: Cluster-specific median rate of change in yearly Chl-a quantiles in shallow and deep zones. The map of the lake shows the areas shallower than 170 m in each cluster.**

## 4 Discussion

The analysis of Chl-a seasonality in Lake Tanganyika reveals a complex and dynamic ecological system that aligns with and expands upon existing studies. It is clear that the seasonality of Chl-a, which includes variations in the location, timing, and intensity of phytoplankton blooms, as well as year-to-year fluctuations, poses challenges for methods that rely solely on field sampling to accurately describe these patterns. Most previous studies did not use remote sensing data and focused on phytoplankton concentrations near the coasts, often in localized areas at either ends of the lake (Plisnier et al., 2018). They could not offer the comprehensive perspective given by the generated maps in Figure 2 that depict the spatial distribution of Chl-a across the whole lake's surface. Our results depict Chl-a seasonality aggregated over a time period that is unprecedented for Lake Tanganyika, whereas other studies were often limited to durations ranging from a single year to a few years (Plisnier et al., 2018).

Figure 2 highlights the difference in Chl-a seasonality between deep and shallow zones. Shallow areas with depths less than 170 m show year-round high Chl-a levels. Notably, such zones are found in different regions around the lake but not along





the whole shoreline. In areas were the lakebed plunges steeply to low depths near the shoreline, Chl-a levels are similar to those observed in pelagic regions. These differences are most evident in April, the period during which interannual median Chl-a concentrations are at their lowest level in all deep areas but remain above 0.3 mg.m$^{-3}$ in shallow areas. This could suggest that a key factor differencing these two types of areas is depth, rather than simply their distance from the shore. Lake Tanganyika could be seen as having two distinct ecosystem areas, each with its own characteristic phytoplankton concentration levels and seasonality. These differences should be attributed to variations in nutrient availability, resulting from the distinct evolutions of water column mixing regimes across different lake zones, as was suggested by Van Bocxlaer et al. (2012).

Bergamino et al. (2010) is the only study that produced maps of Lake Tanganyika's phytoplankton activity, but these maps depicted yearly primary productivity and did not illustrate differences in seasonality between the two zones of varying depths. However, they did delineate areas with the highest Chl-a levels using a clustering method, yielding results comparable to cluster 7 in Figure 3. This cluster groups together different zones, mainly at the north of the lake near the estuary of the Ruzizi river and Bujumbura, the Gulf of Burton, areas near Kalemie, Kigoma and Mpulungu and at the estuary of the Malagarasi river as well as several disparate areas along the coast of Tanzania. Other than their shallow depths, these areas do not share similar spatial contexts: some are located near the estuaries of major rivers, others lie close to urban centres, and some are situated in neither of these settings. A common explanation for the highest Chl-a levels in these areas remains elusive.

Recent studies have, however, made the link between lake depth and trophic levels, although the datasets used are not worldwide, as they only contain data from North America, Europe, and Asia. (Qin et al., 2020; Zhao et al., 2023). These studies show that eutrophication is more prevalent in shallow lakes, where mixing depth and maximum depth are similar, allowing nutrient supply from sediments. In contrast, deep lakes, where mixing depth remains shallower than the mean depth, tend to exhibit more oligotrophic conditions. In this context, Lake Tanganyika appears to exhibit characteristics of both lake types, depending on the depth.

The observed seasonality of Chl-a in pelagic areas aligns closely with the lake's hydrodynamic cycles, as described by Plisnier et al. (2023) and earlier research on pelagic regions of Lake Tanganyika. While Plisnier's hydrodynamic cycle of Lake Tanganyika is subdivided into four distinct periods—two trade wind seasons and two intermonsoon seasons—a three-season framework for Chl-a dynamics emerges from remote sensing observations, as seen in Figure 4.

The first phase begins in April–May, coinciding with the onset of the dry season and southeasterly winds, and extends until August, marking the end of the windy season and the onset of northerly winds. During this period, Chl-a levels rise in the southern part of the lake, peaking at the end of the dry season in August and September, while remaining low in other pelagic regions. This increase is linked to the wind direction which tilts the thermocline, thereby enhancing nutrient availability and promoting phytoplankton blooms. The second phase spans from September to December and is characterized by the reversal of wind directions, a tilting of the thermocline which becomes shallowest in the northern part of the lake and a subsequent dampening oscillation of the thermocline. During this period, Chl-a levels peak in the far north





around September and October. That is followed by a southward progression of elevated Chl-a concentrations. This pattern
aligns with the expected oscillatory behaviour of the thermocline, driving upwellings and subsequent phytoplankton blooms,
especially along the eastern coast of the lake. The highest median concentrations in the south of the lake can be observed in
November, often along the southeast coast of the lake. A pulsed and dampening behaviour is then observed at both ends of
the lake until December, as had been suggested by Plisnier & Coenen (2001). The third phase corresponds to the end of the
rainy season, marked by weak wind dynamics and a sinking thermocline at both ends of the lake. During this phase, Chl-a
levels decrease significantly, reaching their lowest values across all regions of the lake by April.

Multiple sources have predicted or estimated a decline in primary productivity and fish production in Lake Tanganyika
linked to global warming (Bergamino et al., 2010; Cohen et al., 2016; Loiselle et al., 2014; O'Reilly et al., 2003; Tierney et
al., 2010; Verburg, 2006; Verburg et al., 2003). The main driver is the increase in temperature, which increases the stability
of the water column. Figure 5 shows that pelagic Chl-a concentrations have indeed been decreasing between 2002 to 2022, at
rates generally smaller than -0.05 mg.m$^{-3}$ per decade, with an average relative trend of -9% per decade. Considering a
constant conversion factor of 0.505% Chl-a per unit wet weight of phytoplankton biomass (Kasprzak et al., 2008), which
would tend to overestimate the phytoplankton biomass in oligotrophic systems but underestimate it in eutrophic systems, the
obtained overall changes align with previous estimates of trends in productivity estimated by O'Reilly et al., 2003 (-20%
over three decades) and Bergamino et al. (2010) (15% over two decades). If this decline persists beyond the observed 20-
year period, it could have profound consequences for the lake's ecosystem as a whole. While caution is necessary when
estimating the climate's influence on lake fish production, due to the complexity of carbon transfer mechanisms and the
varied outcomes for different phytoplankton groups within the food web (Stenuite et al., 2007), a sustained reduction in
primary productivity will inevitably affect higher trophic levels.

The monthly trends shown in Figure 6 reveal that the overall decrease in Chl-a masks more complex seasonal dynamics.
Indeed, in deep areas, positive trends are observed during the season of highest productivity, while negative trends occur
during the season of lowest productivity. This is confirmed by the analysis in Figure 7, which demonstrates that while the
annual distribution of Chl-a is declining in deep areas, the highest yearly values are actually increasing, potentially indicating
a rise in the frequency of extreme bloom events. However, statistical tests did not detect any significant change in the
frequency or extent of major bloom events. The underlying causes of these intricate deep waters change patterns remain
unclear. Air temperatures increase every month, albeit at different rates, which does not explain the differences in monthly
trends. Variations in wind dynamics could potentially explain these observations; however, there is currently no agreement
on their specific trends. Ground-based wind data are limited, and the reliability of historical wind speed records in the region
remains a subject of debate (Eschenbach, 2004; O'Reilly et al., 2003). Future research should focus on examining the
relative impact of climatic variables on the hydrodynamic cycle in the lake and their respective contributions to the observed
trends in Chl-a.

Coastal zones with depth smaller than 170 m show a mean trend of 0.06 mg.m$^{-3}$ per decade, or 5 % per decade. Figure 6
shows consistent positive coastal monthly trends through the year and Figure 7 shows that most clusters exhibit increases of





Chl-a across all of their distribution. The underlying causes of these changes are uncertain and likely multifactorial. As previously mentioned, these shallow areas are situated in very different contexts. The assumption that industrial or domestic pollution is responsible for higher Chl-a levels does not explain the increased concentrations found in uninhabited shallow coastal regions.

These analyses underscore the pronounced differences between shallow and deep zones, not only in primary productivity levels and seasonality but also in the long-term evolution of phytoplankton concentrations over the past 20 years. This duality suggests that Lake Tanganyika exhibits characteristics of both a shallow lake becoming more eutrophic and a deep lake becoming more oligotrophic, adding to the complexity of its ecological functioning.

## 5 Conclusion

This study provides a comprehensive analysis of the seasonality and long-term trends in surface chlorophyll-a (Chl-a) concentrations across Lake Tanganyika over the past two decades. Our findings highlight the distinct seasonal dynamics of Chl-a, driven by the lake's hydrodynamic processes and regional variations in environmental conditions. The analysis confirms that shallow areas exhibit persistently high Chl-a concentrations year-round, whereas pelagic regions show strong seasonal variability, with peak productivity following wind-driven mixing events.

The long-term trends reveal a complex duality in the lake's ecological evolution. While deep waters are experiencing a general decline in Chl-a concentrations, consistent with previous predictions of reduced primary productivity linked to climate-driven stratification, shallow coastal areas display increasing trends. The contrasting trajectories suggest that Lake Tanganyika is simultaneously exhibiting characteristics of a deep, nutrient-limited system undergoing further oligotrophication and a shallow system experiencing intensified phytoplankton growth. Notably, in deep areas, while overall concentrations are decreasing, the highest yearly values are showing positive trends, suggesting an increase in extreme Chl-a events. These shifts could have cascading effects throughout the lake's trophic network, potentially altering food web dynamics and ecosystem stability.

The observed changes in Chl-a underscore the need for further research into the underlying mechanisms driving these shifts. A better understanding of how climate variability, nutrient dynamics, and anthropogenic pressures interact to shape the lake's primary productivity is crucial. Additionally, assessing the cascading effects on zooplankton and fish populations will be essential for developing effective conservation and management strategies. Given the lake's critical role in supporting surrounding populations, continued monitoring efforts using remote sensing and in situ observations are necessary to guide sustainable resource management in the face of ongoing environmental change.

## Data availability

All data used in this article originate from the freely accessible ESA Lakes_cci dataset (version 2.1) (Carrea et al., 2024).



**Author contribution**

FT, AA, and MV conceptualized the research paper and developed the methodology; FT processed the data, performed statistical analysis and designed the figures; FT wrote the original draft; AA and MV supervised the work and reviewed and
edited the manuscript.

**Competing interests**

Some authors are members of the editorial board of journal Hydrology and Earth System Sciences.

**Acknowledgements**

Generative AI tools were used for sentence rephrasing and programming assistance.

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
