# Peer review of "Leveraging 20 Years of Remote Sensing to Characterize Surface Phytoplankton Seasonality and Long-Term Trends in Lake Tanganyika"

_EGUsphere, 2025_

## Referee Comment (RC2)

**Comment on egusphere-2025-1326**

**1. Section 2.1 Study Site:**

The current scale of the Lake Tanganyika bathymetry map is insufficient to clearly depict variations in water depth and underwater slope from the shoreline toward the central zone. This is particularly relevant for key depth intervals referenced by the author, including the coastal shallow area (<250 m, lines 220 - 221), regions with depths less than 170 m (line 20), and the layer where "phytoplankton biomass is located between depths from 0 to 40 m, with maxima commonly found between 0 - 20 m" (line 106). To improve clarity, it is recommended that several representative east – west cross-sectional depth profiles be included.

**2. Chapter 2: Materials and Methods:**

A comprehensive review suggests that water depth is a primary factor influencing chlorophyll-a (Chl-a), though other variables have not been extensively analyzed. Logically, the analysis should begin by dividing the lake into distinct depth zones, followed by partitioning pixels into grid units. Subsequent steps should involve time-series analysis of these spatial units — covering daily, seasonal, and interannual Chl-a variability — and finally, identification of seven regions with co-varying Chl-a concentrations and their distributional shifts. For instance, after Section 2.2 (Dataset), a new section titled "2.3 Water Depth Zoning and Grid Pixel Unit Division" could be introduced, illustrating unit grids and in-situ Chl-a sampling points on the bathymetric map. This would be followed by Section 2.4 ("Data Application and Interpolation Method"), detailing the interpolation of daily time-step data per pixel. Subsequent sections could cover "2.5 Analysis Methods for Daily, Seasonal, and Interannual Variability of Chlorophyll-a" and "2.6 Partition and Shifts in Chl-a Distribution". It is recommended that this chapter explicitly outline the research framework and methodologies.

**3. Section 2.2 Dataset:**

Does Lake Tanganyika have available data for wind speed, surface water temperature, water turbidity, and in-situ Chl-a? If in-situ Chl-a data exist, have the remote sensing retrieval results been validated against them? Furthermore, the statement that "For Chl-a, 90% of the dataset had an associated fractional uncertainty between 39 and 60%" (lines 42 - 43) raises concerns regarding data reliability and requires further clarification.

**4. Section 2.3 Data Interpolation Method:**

From the context, the missing chlorophyll-a data include some days with completely missing data or incomplete spatial coverage of the lake, which clearly does not include data from the 2012 to 2016 period. However, what method is used to handle the data for lake areas that are not covered? It is suggested that before data interpolation, the chlorophyll data derived from remote sensing interpretation should also be validated against in-situ measurement data.

**5. Sections 2.4, 2.5, and 2.6:**

It is suggested that Sections 2.4, 2.5, and 2.6 be consolidated into a single section titled "Analysis Methods for Seasonal and Interannual Variations of Chlorophyll-a", accompanied by corresponding formulas for calculating seasonal and interannual variability.

**6. Section 2.7 Shifts in Chl-a Distribution: Quantile-Based Trend Analysis:**

This section title is better phrased as "Partition and Shifts in Chl-a Distribution".

**7. Section 3.1 Seasonality of Chl-a Concentrations:**

The author links Chl-a seasonality to water depth, yet Figure 1 is too small to clearly illustrate how depth and slope variations relate to Chl-a concentration. Key coastal shallow areas, such as the Gulf of Burton (line 227), are not marked—consistent with the first comment. While Figure 2 offers a general view of seasonal trends, it is recommended that the April and October/November panels be enlarged to highlight differences between coastal shallow zones and deep-water regions, as well as north – south contrasts. Alternatively, incorporating cross-sectional profiles could better illustrate regional Chl-a dynamics.

**8. Section 3.2 Spatial Clustering for Improved Characterization of Chl-a Variability:**

The analysis describes interannual and seasonal Chl-a variations across seven clusters. Several issues arise:

- The clustering may obscure intra-cluster variability among pixel units, even if differences are small.
- Legend colors in Figure 4 do not match the time-series curves.
- In Figure 4, the seasonal variation curves of chlorophyll-a in Clusters 3 − 6 exhibit a single peak in October, whereas the 75th and 90th percentile daily time series for Clusters 1 and 2 show two distinct peaks—occurring in September and November, respectively. This pattern suggests frequent phytoplankton blooms in the latter clusters, with a decline in October. However, the underlying causes of this differential pattern are not explored. By contrast, the author suggests in the discussion (Lines 367 − 370) that a three-season framework for Chl-a dynamics emerges from remote sensing observations in pelagic regions.

**9. Section 3.2 Overall Patterns of Change in Surface Chl-a:9.The Serial Number of 3.2 is duplicated with the previous section**

This section analyzes decadal trends in annual Chl-a across depth zones (10 m intervals), identifying 170 m as a threshold between positive and negative trends. However, no map compares this 170 m isobath with Chl-a trends. Additionally, the meaning of the p-values in Figure 5c is not explained.

**10. Section 3.3 Monthly Dynamics of Surface Chl-a Trends:**

Only relative trend changes are presented; absolute trend magnitudes and associated p-values are lacking.

**11. Section 3.4 Shifts in Chl-a Distribution: Quantile-Based Trend Analysis:**

Why is the Quantile-Based Trend Analysis from the 25th to 90th percentiles presented as a line graph rather than a bar chart for deep water and shallow water zones? Moreover, both deep water and shallow water zones also obscure the differences among pixels within the same zones.

**12. Chapter 4 Discussion:**

The discussion lacks sufficient supporting data—such as warming trends, wind speed/direction, and hydrodynamic cycles—and the accompanying analysis is inadequate.

Lines 357 – 360: The author mentions that apart from their shallow depths, these areas do not share similar spatial contexts: some are located near the estuaries of major rivers, others lie close to urban centers, and some are situated in neither of these settings. A common explanation for the highest Chl-a levels in these areas remains elusive. From the entire text, aren't the areas with high chlorophyll-a values — including estuarine zones and urban center areas — all located in shallow water regions? Where exactly are the exceptional areas that are not in shallow water? What causes this?

Lines 368 - 370: When mentioning the seasonal variation characteristics of chlorophyll-a in the

deep water area, the author stated: "While Plisnier' s hydrodynamic cycle of Lake Tanganyika is subdivided into four distinct periods—two trade wind seasons and two intermonsoon seasons—a three-season framework for Chl-a dynamics emerges from remote sensing observations, as seen in Figure 4." However, Figure 4 indicates that except for the 7th shallow water cluster, the seasonal variations of chlorophyll-a in the other several clusters do not show results consistent with a three-season framework.

Lines 385 – 397: The text discusses the issue of declining primary productivity and fish yields caused by rising temperatures. However, Figure 5 reflects the long-term changes in chlorophyll a. So, what are the influencing factors of primary productivity and fish yields, and do they show a positive correlation with chlorophyll a? Moreover, the article does not cite enough data on the warming temperature trend to support this inference.